# Probiotics as a Treatment for “Metabolic Depression”? A Rationale for Future Studies

**DOI:** 10.3390/ph14040384

**Published:** 2021-04-20

**Authors:** Oliwia Gawlik-Kotelnicka, Dominik Strzelecki

**Affiliations:** Department of Affective and Psychotic Disorders, Medical University of Lodz, 90-419 Lodz, Poland; dominik.strzelecki@umed.lodz.pl

**Keywords:** depression, obesity, metabolic syndrome, probiotics, microbiota

## Abstract

Depression and metabolic diseases often coexist, having several features in common, e.g., chronic low-grade inflammation and intestinal dysbiosis. Different microbiota interventions have been proposed to be used as a treatment for these disorders. In the paper, we review the efficacy of probiotics in depressive disorders, obesity, metabolic syndrome and its liver equivalent based on the published experimental studies, clinical trials and meta-analyses. Probiotics seem to be effective in reducing depressive symptoms when administered in addition to antidepressants. Additionally, probiotics intake may ameliorate some of the clinical components of metabolic diseases. However, standardized methodology regarding probiotics use in clinical trials has not been established yet. In this narrative review, we discuss current knowledge on the recently used methodology with its strengths and limitations and propose criteria that may be implemented to create a new study of the effectiveness of probiotics in depressive disorders comorbid with metabolic abnormalities. We put across our choice on type of study population, probiotics genus, strains, dosages and formulations, intervention period, as well as primary and secondary outcome measures.

## 1. Introduction

Depressive disorders and metabolic syndrome (MetS) are two of the most common and disabling civilization diseases [1]. Depression is a major risk factor for everyday disablement or suicide, and MetS may lead to cardiovascular diseases (CVD) and type 2 diabetes mellitus (2DM). Additionally, depressive disorders are often comorbid with MetS increasing mortality risks [2,3]. A former meta-analysis [4] showed that individuals with depression suffered many metabolic abnormalities and had a 1.5 times higher odds of having MetS; prevalence of MetS in depressive subjects accounted for 30%. A very recent meta-analysis of 49 studies confirmed a significant relationship between depression and MetS (the pooled Odds Ratio of MetS in patients with depression was 1.48 % in 31 cross-sectional studies and the pooled Risk Ratio 1.38 in cohort studies) [5]. Moreover, data from an 808-person sample with a current diagnosis of depression did demonstrate that persons with atypical depression had significantly higher levels of inflammatory markers, body mass index (BMI), waist circumference (WC) and triglycerides (TG), and lower high-density lipoprotein cholesterol (HDL-C) than subjects with melancholic depression [6]. Additionally, in a large, nationally representative sample, it was found that both obesity and MetS were associated with significant depressive symptoms independent of each other, and that participants with both conditions had the highest rate of depression compared to the other groups [7]. Another state, strongly associated with both depressive and MetS issues, is non-alcoholic fatty liver disease (NAFLD). NAFLD is a multisystem disease that is considered the hepatic manifestation of MetS and is characterized by excessive hepatic fat accumulation [8]. To underline its nature, recently it has been proposed to rename the syndrome as metabolic-associated fatty liver disease (MAFLD) [9]. Moreover, similarly to MetS, most deaths among NAFLD patients are attributable to CVD [10]. Altogether, the term "metabolic syndrome type II" was proposed in 2007 as a neuropsychiatric syndrome in which alterations in metabolic networks are a defining pathophysiological component [11]. In this narrative review, we use the term “metabolic depression” to describe depressive disorders comorbid with obesity, MetS or NAFLD.

Although the exact mechanisms underlying this association between depression, MetS and its liver equivalent are poorly known, several hypotheses have been proposed. First of all, antipsychotics augmentation in depression is associated with significantly higher metabolic abnormalities prevalence and increased mortality risk [12]. Secondly, lifestyle factors, especially diet and physical activity, explained more than one fifth of the association between depressive and metabolic disorders [13]. Remaining factors participating in the coexistence of depressive and metabolic diseases are poorly known; however, a possible pathophysiological overlap has been proposed [14]. 

The hypothalamic–pituitary–adrenal (HPA) axis dysregulation with final hypothalamic inflammation is one of the factors depression and MetS have in common [15]. In typical depression, the axis is upregulated with excess release of cortisol [16]. However, atypical depression is associated with a hypofunction of the HPA axis. In metabolic disorders, rather, neuroendocrine dysregulatory mechanisms are involved [17]. Furthermore, studies have uncovered that both depression and metabolic diseases are associated with chronic, low-grade inflammation [7,18,19]. There is also a number of studies that oxidative stress (OxS) may be involved in the pathophysiology of metabolic [20,21] as well as depressive disorders [22,23,24].

In recent years, there has been much interest in the role of microbiota changes (dysbiosis) in the development of chronic inflammation and civilization diseases [25]. Intestinal microbiota play an important role in regulating the brain functions of the host through the gut–brain axis (GBA) [26]. The gut dysbiosis has been found to play a significant role in the occurrence of mood and anxiety disorders [27,28,29,30]. Additionally, there is scientific data on the participation of the gut microbiota dysfunction in the onset of obesity-related disorders [31,32], as well as the pathogenesis of NAFLD [33,34]. Additionally, there is more and more evidence that an aberrant gut microbiota may lead to chronic inflammation [35,36] and OxS exacerbation in tissues [37] so that may serve as a link between MDs, depression and dysbiosis.

Elucidating these mechanisms linking metabolic diseases, depression, HPA dysregulation, CLGI, OxS and dysbiosis could generate potential new therapeutic means or patient-specific strategies to combat both metabolic and depressive disorders, e.g., microbiota interventions.

## 2. Microbiota Interventions

Several possible interventions on our microbiota have been described in the literature [38,39]. Of them, proper diet and other lifestyle factors are established ways of alleviating symptoms of metabolic diseases, as well as an adjunctive method in the treatment of depression [40,41,42]. Results of research on prebiotics have suggested favourable outcomes in metabolic disorders: decreased fasting glucose (fGlc), improved insulin sensitivity and lipid profile, reduced inflammation including neuroinflammation; however, when investigated as isolated therapies, prebiotics did not impact outcomes’ measures of depression or anxiety [43,44,45,46,47]. Interestingly, there appears to be some evidence for the treatment of psychiatric and metabolic disorders through fecal microbiota transplantation (FMT); additionally, FMT did have the potential to reduce small intestinal permeability in patients with NAFLD [48,49,50]. Another clinical intervention involving microbiota is the use of postbiotics (metabiotics). The most common types of postbiotics are microbial metabolites: short-chain fatty acids (SCFAs), peptides, enzymes, teichoic acids, and vitamins [51]. Among others, it was shown that lipoteichoic acid from B. animalis strain was responsible for its fat-reducing properties [52] and SCFAs administration alleviated symptoms of depression in mice [53]. SCFAs may also be used as a candidate agent in the prevention and treatment of obesity by inducing thermogenesis in brown adipose tissue and browning in white adipose tissue [54].

The intervention on our microbiota being the core topic of this review is supplementation with probiotics. The most accepted definition of probiotics is “live microorganisms, that when consumed in adequate amounts, confer a health effect on the host” [55]. 

### 2.1. Probiotics in Experimental Studies

Experimental studies have uncovered psychoactive properties of several probiotics formulations in different models of GBA dysfunction as well as in animal models of mental health problems, including anxiety and depression. Lactobacillus (L.) rhamnosus JB-1 was shown to prevent some of the antibiotic-induced alterations in rodents, e.g., impaired anxiety and social behaviours, as well as increased levels of aggression [56]. Moreover, Bifidobacterium (B.) longum subsp. infantis E41 and B. breve M2CF22M7 were shown to have an antidepressant effect in mice [57]. Importantly, probiotic pretreatment (L. helveticus R0052 + B. longum R0175) significantly alleviated hippocampal apoptosis induced by lipopolysaccharide in rats, suggesting that this probiotic could play a role in some neurodegenerative conditions [58,59]. Furthermore, not only stress-induced anxiety or depressive-like behaviours, but also cognitive deficits along with the reduced level of brain-derived neurotropic factor (BDNF) were alleviated by L. plantarum WLPL04 [60]. Additionally, overall, probiotics reduced anxiety-like behaviour in animals, but only among diseased ones [61]. Furthermore, the comparison of efficacy of twelve candidate probiotic strains of Bifidobacterium and Lactobacillus in a dose of 1 × 10^9^ CFU/day on chronically stressed mice showed that L. paracasei Lpc-37, L. plantarum LP12407, L. plantarum LP12418 and L. plantarum LP12151 prevented stress-associated anxiety and depression-related behaviours [62] adding data on probiotics strain-dependence. On the contrary, L. helveticus R0052 and B. longum R0175 promoted an anti-inflammatory profile but not reductions in behavioural responses to social stress in hamsters [63].

As regards to metabolic abnormalities, L. fermentum alleviated inflammation and intestinal barrier integrity dysfunction, as well as improved insulin sensitivity in diet- and streptozotocin-induced diabetes in rats [64]. In other studies, L. fermentum CECT5716 exerted anti-obesity effects, associated with its anti-inflammatory properties and ameliorated endothelial dysfunction and gut dysbiosis in a model of high-fat diet (HFD)-induced obesity in mice [65]; and L. fermentum CRL1446 administration improved adiposity index, inflammatory, oxidative, glucose and lipid profiles and favourably modulated intestinal microbiota in mice with MetS [66]. Similarly, the administration of B. animalis subsp. lactis BB-12 and L. plantarum 299v to diet-induced obesity and MetS in rabbits demonstrated favourable effects on several metabolic abnormalities [67]. In concordance with that, the administration of L. acidophilus probiotic in rats with MetS caused by fructose reduced insulin resistance (IR) [68]. As regards MetS-associated OxS and liver function biomarkers, L. pentosus GSSK2 and L. plantarum GS26A were equally effective in HFD-induced MetS in rats [69]. Additionally, in preclinical studies, a number of probiotic genus, such as Bifidobacterium, Lactobacillus, or Bacillus, have shown beneficial effects in rodent models of NAFLD [70,71,72,73] and probiotics are now considered as a potential therapy for human NAFLD.

Interestingly, intervention with GABA-producing L. brevis DPC6108 and L. brevis DSM32386 improved both metabolic abnormalities and depressive-like behaviour associated with MetS in mice [74].

Altogether, the results have provided an experimental basis for the prophylactic and adjunct therapeutic application of probiotics on depression comorbid with lifestyle-related disorders such as obesity, MetS and its aftereffects. Furthermore, the experimental studies results may serve as an indicator for selecting specific strains and dosages of probiotics. However, animals’ results cannot always be transferred to human studies (Table 1).

### 2.2. Probiotics in Human Studies

#### 2.2.1. Systematic Reviews and Meta-Analyses of Randomized Clinical Trials (RCTs) of Probiotics Interventions 

Systematic reviews and meta-analyses of human trials using probiotics demonstrated their usefulness in depressive or, less consistently, in anxiety outcome measures [46,47,61,75,76,77,78,79,80,81,82,83,84,85,86,87]. It has been suggested that the microorganisms can form a new group of drugs named “psychobiotics” [88].

The first meta-analysis in the field showed that probiotics significantly decreased the depression scale score. However, among five included studies only one had investigated patients with major depressive disorder (MDD) and four the healthy population. Regarding patients’ age, probiotics had an effect on the population aged under 60, but not on people aged over 65 [81]. In another study, there was no significant difference in mood between the probiotic and placebo group post-intervention; nonetheless, a subgroup analysis of studies conducted in healthy versus depressed subjects showed significant improvements in the mild-to-moderate depression group [82]. Similarly, a beneficial effect of probiotics on depressive symptoms when administered to clinical/medical samples has been shown in several meta-analyses [46,75,79,80,85,87], with a larger effect observed for psychiatric, especially MDD, samples, for longer intervention periods (more than a month) and with multiple strains formulations [46,85,87]. Moreover, a significant reduction in Hamilton Depression Rating Scale (HDRS), C-reactive protein (CRP), interleukin (IL)-10 and malondialdehyde (MDA) levels was found in patients with psychiatric disorders after probiotics supplementation [75]. According to the newest meta-analysis of RCTs in clinical depression samples, probiotics are effective in reducing depressive symptoms when administered in addition to antidepressants; however, they are not significantly advantageous over placebo when used as stand-alone treatment [78].

Additionally, with regard to anxiety, the meta-analyses revealed no significant difference between probiotics and placebo in alleviating anxiety symptoms, and did not differentially affect clinical and healthy human samples [61,76,79].

The recent systematic review and meta-analysis has confirmed the effectiveness of probiotics on the amelioration of anthropometric measures (body mass index (BMI), waist circumference (WC) and hip circumference (HC)) of overweight and obese patients with related metabolic diseases [89]. This is partly in agreement with several previous meta-analyses in obese subjects [90,91,92,93,94,95,96,97]; however, the studies lack consistency [98,99,100]. The meta-analysis assessing the effect of probiotics on risk factors of cardiometabolic diseases in healthy people revealed reduced BMI and WC and, additionally, total cholesterol (TC) in overweight subjects. Most of the genera were Lactobacillus and Bifidobacterium, the mean time of probiotic administration was 67 days and the daily probiotic dose varied between 10^6^ and 10^10^ colony-forming units (CFU)/gram [101]. Another meta-analysis confirmed that the improvements in metabolic variables were mostly observed with Bifidobacterium and Lactobacillus genera, adding data on favourable effects of Streptococcus salivarius subsp. thermophilus and mixtures of probiotic strains [93]. Contrary to the previously cited results, probiotic supplements did not have favourable effects in overweight or obese children and adolescents [102], nor were they superior to placebo in overweight or obese pregnant women for the prevention of gestational diabetes mellitus [103]. Additionally, probiotics may have no-to-minor effect regarding weight loss post bariatric surgery [104].

Furthermore, it has been summarized that probiotic intake may ameliorate some of the clinical components of MetS but the results are inconclusive [93,96,97,98,105,106,107]. Analysis of patients with CVD risk factors revealed favourable effects of probiotics with longer duration of treatment (>1.5 months), higher dosage of probiotics (>1.0 × 10^9^ CFU), diabetic patients and female populations [97]. Another systematic review of 6 RCTs of probiotic strains for modulating obesity-related microbiota dysbiosis showed that Lactobacillus genus was administered twice as often than *Bifidobacterium*. Additionally, the daily dose varied from 1×10^8^ to 1.35 × 10^15^ CFU/day, and the time of administration varied from 4 to 24 weeks [108].

Importantly, levels of some inflammatory biomarkers associated with MetS were found to be decreased after probiotics treatment, e.g., serum CRP, the soluble vascular cell adhesion molecule 1 (sVCAM-1), IL-6, tumour necrosis factor α (TNF-α), vascular endothelial growth factor (VEGF), and thrombomodulin [100,107,109]. Similarly, an improvement in clinical features as well as in OxS biomarkers was found in type 2 diabetes mellitus (2DM) patients [110] or polycystic ovary syndrome (PCOS) patients [111] after probiotic supplementation. 

Meanwhile, clinical research on probiotics conducted so far has found positive results in NAFLD subjects. Probiotics in recent systematic reviews and meta-analyses were shown to be superior to placebo regarding several anthropometric and laboratory metabolic parameters, including serum aspartate aminotransferase (AST) and alanine aminotransferase (ALT) in both adult and paediatric patients [93,95,112], as well as inflammation markers [112,113] in NAFLD patients. 

To summarize, most systematic reviews and meta-analyses have demonstrated some benefit with respect to body weight loss and BMI among adult participants; however, these changes were rather discreet and of little significance for overall health status. The results concerning other criteria of MetS are inconclusive. However, there seems to be an agreement in term of probiotics efficacy toward NAFLD severity.

Table 2 shows a brief summary of meta-analyses of RCTs with probiotics in the field of depressive and anxiety symptoms, metabolic parameters, including obesity, MetS, and its liver equivalent published in recent years.

To conclude, it seems reasonable that current probiotics formulations should only be used as a complementary treatment for both depressive and metabolic disorders. Importantly, probiotics use is associated with minor or no adverse events, thus their supplementation might be worth considering. The question arises as to whether comorbidity of depressive disorders with obesity/MetS/NAFLD may serve as a specific indication for probiotics therapy. 

Additionally, one must bear in mind that, as opposed to the meta-analyses conducted in pharmacologic agents, meta-analyses conducted in nutritional interventions are not always the best method for extracting relevant information, due to the heterogeneity of formulations and protocols. 

To summarize, the majority of preclinical studies and meta-analyses of clinical observations support further studies of probiotics in the treatment of depressive disorders, obesity and metabolic disorders. Considering good safety and tolerability profile, it seems worthwhile to investigate the most efficacious probiotics regimens, including interventional timing, treatment duration, strain-dependency, dosage, etc. 

#### 2.2.2. Key Features of RCTs with Probiotics

In past years, dozens of RCTs trials have been carried out to compare probiotics, including Bifidobacterium and Lactobacillus, with placebo in metabolic disorders, and, not as many, in the depressive disorders population. Table 3 presents all RCTs with probiotics that have been published so far in the depressed population and selected RCTs results in obesity, MetS and the NAFLD population.

In depressive and anxiety clinical populations research, L. acidophilus has been the most often studied probiotics species so far [114,119,127,129,133,137]. L. casei and B. bifidum are runners-up. However, it is worth noting that there have been only 10 such RCTs results published and most of them used a combination of different bacterial genera and species. Additionally, the dosages of probiotics varied from 2 × 10^9^ to 2 × 10^10^ CFU per day, and the form of administration was various (cheese, yogurt, milk or probiotic capsule). Moreover, the time of intervention varied from 4 weeks to 90 days. Regarding the subjects included, one study only recruited pregnant women [137], the other depression/anxiety in irritable bowel syndrome (IBS) [135], two treatment-resistant depression [133] and the rest of the trials included patients diagnosed as being generally depressed or having MDD. The sample size varied from 12 to 82 subjects. Furthermore, probiotics supplementation has been used definitely more often as an add-on then stand-alone treatment for depression/anxiety. Moreover, the effects of probiotics have yet to be tested in a clinical sample of treatment-naïve depressed patients. Additionally, some of the studies used a prebiotic concurrently with probiotics (symbiotic formulation); recently, it has been shown that adding inulin (a prebiotic) to probiotics improved psychological and inflammatory outcomes more effectively than two supplements separately [139]. Finally, the findings have indicated that probiotic supplementation is safe and well-tolerated.

It is worth mentioning that several strains of probiotics may be especially promising for the complementary treatment of metabolic disorders. L. gasseri, originated from human breast milk, has been proven to reduce several anthropometric parameters in abdominal obesity subjects in doses as low as 2 × 10^8^/day [117]. Interestingly, L. gasseri BNR17 received approval as a treatment for body fat reduction in South Korea [140]. Akkermansia muciniphila has been shown to be negatively correlated with obesity and its supplementation ameliorated some metabolic parameters [118]. In several studies, L. acidophilus and B. lactis given together have been shown to reduce not only obesity and MetS parameters, but also some inflammation and OxS markers connected with the syndrome [106,115,120,141]. 

These studies have provided necessary pilot data regarding the efficacy and safety profiles of probiotics in clinical practice and paved the way for more elaborate probiotic pharmacotherapies in the future.

Bifidobacterium and Lactobacillus strains are still the most widely used probiotics in civilization diseases complementary treatment research. Next-generation probiotics, such as Faecalibacterium prausnitzii, Akkermansia muciniphila, or Clostridia strains, supplements await further development.

### 2.3. Conclusions

To sum up, due to the small number and scale of studies and heterogeneity of population, probiotic strains and genus, administered doses, the period of the interventions, and of endpoints it is hard to come to firm conclusions. 

Firstly, probiotics are considered evidence-based treatment for antibiotic- or Clostridium difficile-associated diarrhoea and respiratory tract infections, but not for depression nor metabolic diseases [142]. Examples of psychobiotic strains that were found to be somehow effective to counteract affective and anxiety symptoms include: L. fermentum NS8 and NS9, L. casei Shirota, L. gasseri OLL2809, L. rhamnosus JB-1, L. helveticus Rosell -52, L. acidophilus W37, L. brevis W63, L. lactis W19 and W58, B. longum Rosell-175, B. longum NCC3001, B. longum 1714, B. bifidum W23, B. lactis W52, L. plantarum 299v [143]. However, there is an urgent need for a standardised methodology in this area to determine the exact type of probiotic administered to the “metabolic depression” subpopulation.

Furthermore, it is necessary to consider possible effects of the co-administration of different types of probiotics or a probiotic and prebiotic administered as it might influence the effectiveness of the first or the latter.

Additionally, the exact dose of probiotics should be determined, as higher doses may cause adverse effects and lower ones may be ineffective. Moreover, the formulations used should be thoroughly described as different formulations may account for different absorbance rates and thus different results.

Also, the time of intervention influence outcomes of the treatment. It seems that it is reasonable to supplement the probiotic formulation for at least 6–8 weeks to reduce the depression scale score; however, it may not be enough to affect some metabolic parameters. Durability of beneficial effects is also controversial as discontinuation of the treatment may result in loss of its effectiveness after some time. Follow-up visits may explain this question.

Secondly, as regard populations studied, there is a shortage of studies in humans, and preclinical studies may not reflect human physiology. The included populations are heterogenous and this may affect the results of a trial. Moreover, sample sizes of the trials are small, causing reduced statistical power. The major drawback of current methodology in probiotics trials is a lack of personalization of treatment; each person has a unique microbiota and may need an individualized approach. Additionally, it should be underlined that different individuals can have taxonomically varied but functionally similar microbiota, which makes a functional rather than taxonomic approach to creating probiotics formulations more important [144]. One of the potential microbial function biomarkers are short-chain fatty acids (SCFAs) that are the most representative metabolites of fiber anaerobic fermentation [145]. Interestingly, a depletion of SCFAs was reported in MDD patients [146], and SCFAs can play an important role in regulating metabolic and cardiovascular health [33,147]. Therefore, SCFAs-producing bacteria may become an interesting target as a potential treatment of metabolic depression. 

Finally, regarding outcome measures, the methodology needs standardization. Clinical diagnosis and professional-assessed psychometric scales are a good option in psychiatric population. When it comes to metabolic parameters, it would be advisable to use WC rather than BMI and incorporate MetS criteria into trial secondary outcome measures.

Current treatments for both depression and metabolic diseases remain suboptimal for many patients, making improvements and advances in the intervention options in great demand. Whilst microbiota interventions may have benefits for some individuals, possibly those with comorbid obesity/MetS, evidence-based probiotic treatment awaits development. 

We suggest that the mechanisms of action of probiotics in relation to "metabolic depression" are: primarily decreasing chronic inflammation as well as pro-oxidative states, and additionally balancing HPA dysfunction.

## 3. Practical Applications

To evaluate psychobiotic potential of a microorganism or microbial formulations according to a general methodology proposed by del Toro-Barbosa [148], there are 4 steps in the procedure: 1. Formulation, 2. In vitro tests in bacteria or mammalian cells, 3. Pre-clinical tests in murine models and 4. Clinical studies. As the abovementioned formulations have been extensively studied up to the 3rd step and there is not enough evidence for their efficacy from clinical studies, it is crucial to plan appropriate clinical study protocols. 

Newly constructed study protocol could include adult patients diagnosed with depressive disorders with/without the MetS and/or NAFLD. The protocol might include psychometric and anthropometric parameters, as well as laboratory tests (MetS criteria, indicators of liver fibrosis, fecal microbiota analysis and possibly biological markers such as cortisol, inflammation and OxS parameters and brain imaging studies). The study could enable to establish a subpopulation of patients sensitive to microbiota interventions, especially probiotics, as an add-on therapy, as well as to determine potential biomarkers of therapeutic efficacy of probiotics. 

As for psychobiotics choice, it is well-known that the probiotic effects are strongly strain-dependent and there is a consensus that a mixture of collaborating microbes would be more beneficial than a single strain. Generally, strains from genera Lactobacillus and Bifidobacterium in combination are worth studying, e.g., well-studied probiotic mixture of bacterial strains consists of L. helveticus R0052 and B. longum R0175 [59,116,149,150,151,152,153,154,155,156]. Recently, an open-label pilot study has been published that adds evidence to the antidepressant potential of this probiotic formulation [157]. Additionally, a RCT protocol of a study assessing antidepressant properties of the formulation in the context of MetS comorbidity has been published [158].

As there are several different definitions of depression applied in clinical and research practice [159] and, according to upcoming ICD-11, depressive disorders include not only MDD, but also dysthymic and mixed depressive and anxiety disorder (MDAD), underlying their impact on patients’ everyday functioning and quality of life, as well as importance in primary care settings [160,161], it would be valuable to incorporate the whole category into probiotics trials.

Additionally, it is worth remembering that there are many factors impacting microbiota function and composition, e.g., the diet, [39,162,163,164,165,166,167,168,169,170,171,172,173,174] and they should be assessed along the study process.

Proposed features of a new randomized clinical trial protocol of probiotics efficacy in depressive patients with metabolic abnormalities are summarized in Table 4.

Based on the above, we have just registered a randomized clinical trial on the influence of probiotic supplementation on depressive symptoms, inflammation, oxidative stress and fecal microbiota in depressed patients depending on MetS comorbidity (ClinicalTrials.gov identifier: NCT04756544) [158].

Overall, the effectiveness of probiotics in the prevention and treatment of depression, obesity and metabolic diseases remains to be elucidated in future large-scale studies in clinical populations. Additionally, the ideal mixture of probiotic strains, dose, the duration of supplementation, and the durability of beneficial effects are to be established. 

## Figures and Tables

**Table 1 pharmaceuticals-14-00384-t001:** Summary of the most important data from pre-clinical studies on probiotics in some mental health and metabolic disorders models.

Data	Mental Health Problems Models	Metabolic Disorders Models
Main clinical features findings	Prevention of anxiety and depression, antidepressant and antianxiety effect, alleviation of cognitive deficits	Anti-obesity effect
Main laboratory findings	Prevention of hippocampal apoptosis, reduction of brain-derived neurotropic factor (BDNF) level, promotion of an anti-inflammatory profile	Alleviation of inflammation, oxidation, endothelial dysfunction and intestinal barrier integrity dysfunction, improvement in insulin sensitivity, glucose and lipid profiles, liver function biomarkers
Commonly studied probiotics	Lactobacillus (e.g., L. plantarum) and Bifidobacterium genera	Lactobacillus (e.g., L. fermentum) and Bifidobacterium genera

**Table 2 pharmaceuticals-14-00384-t002:** Meta-analyses of RCTs with probiotics efficacy towards depressive and anxiety symptoms and recent meta-analyses of RCTs with probiotics efficacy toward metabolic health parameters.

**Depressive and Anxiety Symptoms**			
Amirani et al. 2020 [75]	12 RCTs	656 subjects	Reduced the HDRS score by 9.60. Reduced CRP by 1.59 mg/L, TNF-α by 0.12 pg/mL, and MDA by 0.38 μmol/L.
Chao et al. 2020 [79]	10 RCTs	685 subjects	Reduced the depression scale score by 0.47. No significant impact on anxiety symptoms.
Goh et al. 2019 [85]	19 RCTs	1901 subjects	Reduced the depression scale score by 0.31.
Huang et al. 2016 [81]	5 RCTs	365 adult subjects	Reduced the depression scale score by 0.30.
Liu et al. 2018 [76]	12 RCTs	1551 subjects	No significant impact on anxiety symptoms.
Liu et al. 2019 [46]	29 RCTs	?	Reduced the depression scale score by 0.24 and the anxiety scale score by 0.10.
Ng et al. 2018 [82]	10 RCTs	1349 subjects	Reduced the depression scale score by 0.684 in mild/moderate depression. No significant difference in mood overall (healthy and clinical population).
Nikolova et al 2021 [78]	7 RCTs	404 subjects	Reduced the depression scale score by 0.83 as an add-on. No significant impact as a standalone treatment.
Nikolova et al. 2019 [80]	3 RCTs	229 subjects	Reduced the depression scale score by 1.371.
Reis et al. 2018 [61]	14 RCTs	1527 subjects	No significant impact on anxiety symptoms.
Sanada et al. 2020 [87]	6 RCTs	302 subjects	Reduced the depression scale score by 1.62.
**Obesity, MetS, NAFLD, and metabolic parameters in healthy subjects**			
Borgeraas H et al. 2018 [92]	15 RCTs	957 subjects	Reduced BW by 0.60 kg, BMI by 0.27 kg/m^2^ and fat percentage by 0.60%
Chatzakis et al. 2019 [103]	5 RCTs	1235 overweight or obese pregnant women	No significant impact on GDM risk, nor gestational weight gain.
Companys et al. 2020 [96]	52 RCTs	Overweight/obese/hypercholesterolemia/MetS subjects	Reduced BW, BMI, WC, BFP. Improved lipids profile.
Dixon et al. 2020 [97]	34 RCTs	2177 hypertension, obesity, CVD, MetS, T2D or hypercholesterolaemia subjects	Reduced SBP by 1.31 mmHg, DBP by 1.87 mmHg, TC by 6.05 mg/dL, LDL-C by 8.77 mg/dL, fGlc by 4.92 mg/dL, HbA1C by 0.18%, BMI by 0.31 kg/m^2^. Increased HDL-C by 1.05 mg/dL. No significant effect on TG.
Dong et al. 2019 [98]	18 RCTs	1544 subjects	Reduced BFP by 0.3% and LDL-c by 0.18 mg/dL; No significant differences of BMI, BFM, WC, HC, WHR, SBP, DBP, fGlc, fasting insulin, TC, HDL-C, HbA1c, or TG.
Kazemi et al. 2019 [100]	29 RCTs	Metabolic disorders (e.g., NAFLD and MetS) subjects	No significant impact on BMI. Reduced CRP by 0.32 mg/L.
Koutnikova H et al. 2019 [93]	111 RCTs	6826 (e.g., obese, NAFLD) subjects	Reduced body weight by 0.94 kg, BMI by 0.55 kg/m^2^, WC by 1.31 cm, BFM by 0.96 kg, and visceral adipose tissue mass by 6.30 cm^2^
Kunnackal et al. 2018 [91]	22 RCTs	?	Reduced BW by 0.65 kg, BFM by 0.94 kg and BMI by 0.33 kg/m^2^
Mohammadi et al. 2019 [102]	9 RCTs	410 overweight or obese children and adolescents	No significant changes in BMI, WC, BW, BFM, fGlc and lipid profiles.
Pan et al. 2020 [113]	11 RCTs	NAFLD subjects	Reduced TNF-α by 0.52 pg/mL and CRP by 0.62 mg/L.
Park S et al. 2015 [99]	4 RCTs	449 adult subjects	No significant effect on body weight and BMI
Perna et al. 2021 [89]	20 RCTs	1411 subjects	Reduced BMI by 0.73 kg/m^2^, WC by 0.71 cm and HC by 0.73 cm. No significant effect on body weight.
Skonieczna-Żydecka et al. 2020 [101]	61 RCTs	5422 healthy subjects (including overweight/obese ones)	Reduced BMI by 0.45 kg/m^2^, WC by 1.21 cm in healthy persons. Reduced TC in overweight/obese subjects. No significant impact on carbohydrate and lipid metabolism
Swierz et al. 2020 [104]	5 RCTs	Morbid obesity undergoing bariatric surgery subjects	No significant effect on body weight.
Tang et al. 2019 [112]	18 RCTs	NAFLD subjects	Reduced weight by 2.31 kg, and BMI by 1.08 kg/m^2^. Reduced ALT by 7.22 U/L, AST by 7.22 U/L, AP by 25.87 U/L, GTP by 5.76 U/L. Reduced TC by 0.73, LDL-C by 0.54, TG by 0.36 mg/dL. Reduced fGlc by 4.45 mg/dL, insulin by 0.63 µIU/mL. Reduced TNF-α by 0.62 pg/mL, and leptin by 1.14 ng/mL.
Wang ZB et al. 2019 [94]	12 RCTs	821 adult subjects	Reduced BW by 0.55 kg, BMI by 0.30 kg/m^2^, WC by 1.20 cm, BFM by 0.91 kg, and BFP by 0.92%
Xiao et al. 2019 [95]	28 RCTs	1555 NAFLD subjects	Reduced BMI by 1.46 kg/m^2^, ALT by 13.40 U/L, AST by 13.54 U/L, GTP by 9.88 U/L, insulin by 1.32 𝜇IU/mL, and TC by 15.38 mg/dL; No significant effect on fGlc, lipid profile or TNFα.
Zhang Q et al. 2016 [90]	25 RCTs	1931 adult subjects	Reduced BW by 0.59 kg and BMI by 0.49 kg/m^2^

Abbreviations: ALT: alanine transaminase; AP: alkaline phosphatase; AST: aspartate transaminase; BMI: body mass index; BFM: body fat mass; BFP: body fat percentage; CRP: C-reactive protein; CVD: cardiovascular disease; DBP: diastolic blood pressure; fGlc: fasting glucose; GDM: gestational diabetes mellitus; GTP: gamma-glutamyl transferase; HbA1c: haemoglobin A1c; HC: hip circumference; HDL-C: high-density lipoprotein cholesterol; HDRS: Hamilton Depression Rating Scale; LDL-C: low-density lipoprotein cholesterol; MDA: malondialdehyde; MetS: metabolic syndrome; NAFLD: non-alcoholic fatty liver disease; SBP: systolic blood pressure; TNF-α: tumour necrosis factor α; RCT: randomized clinical trial; T2D: type 2 diabetes mellitus; TC: total cholesterol; TG: triglycerides; WC: waist circumference; WHR: waist-to-hip ratio.

**Table 3 pharmaceuticals-14-00384-t003:** The selected recent randomized clinical trials with probiotics formulations in the field of depressive and metabolic disorders.

Clinical trials: size; type; duration; probiotic formulation	**Depression**	**Obesity, MetS and NAFLD**
	Akkasheh, 2016 [114]: MDD; 40, add-on, 8 weeks; L. acidophilus (2 × 10^9^), L. casei (2 × 10^9^), B. bifidum (2 × 10^9^).	Szulińska 2018 [106,115]: 81; 12 weeks; B. bifidum W23, B. lactis W51, B. lactis W52, L. acidophilus W37, L. brevis W63, L. casei W56, L. salivarius W24, L. lactis W19, L. lactis W58, lyophilisate powder, low dose (2.5 × 10^9^ CFU/day) or high dose (1 × 10^10^ CFU/day)
	Romijn et al. 2017 [116]: Depressive and anxiety disorders; 79; standalone; 8 weeks; L. helveticus R0052, B. longum R0175 (≥3 × 10^9^/day)	Kadooka et al. 2013 [117]: Obesity; 210; 12 weeks; L. gasseri SBT2055 2 × 10^8^/day
	Kazemi, 2019 [109]: MDD; 74; add-on; 8 weeks; L. helveticus + B. longum; ≥10 × 10^9^/day.	Depommier et al. 2019 [118]: overweight/obese insulin-resistant subjects; 40; 3 months; Akkermansia muciniphila 10^10^ CFU/day.
	Ghorbani, 2018 [119]: MDD; 40; add-on; 6 weeks; L. casei 3 × 10^8^, L. acidofilus 2 × 10^8^, L. bulgaricus 2 × 10^9^, L. rhamnosus 3 × 10^8^, B. breve 2 × 10^8^, B. longum 1 × 10^9^, Streptococcus thermophilus 3 × 10^8^ (plus prebiotic).	Rezazadeh 2019 [120]: 44; 8 weeks; yogurt containing 6.45 × 10^6^ CFU/g of L. acidophilus La5 and 4.94 × 10^6^ of B. lactis Bb12
	Miyaoka,2018 [121]: TRD; 40; add-on; 8 weeks; C. butyricum, 60 mg daily.	Leber 2012 [122], Tripolt 2013 [123], Stadlbauer 2012 [124]: 28; add-on; 12 weeks; L. casei Shirota, milk (65 mL bottles × 3/day) 10^8^ cells/mL.
	Rudzki, 2019 [125]: MDD; 60; add-on; 8 weeks; L. plantarum (10 × 10^9^).	Sharafedtinov 2013 [126]: 40; 3 weeks; add-on; L. plantarum TENSIA, cheese (50 g/day) 1.5 × 10^11^ CFU/g.
	Chahwan, 2019 [127]: depressive disorders; 71; stand-alone; 8 weeks; B. bifidum, B. lactis W51 & W52, L. acidophilus, L. brevis, L. casei, L. salivarius, and Lactococcus lactis W19 & W58; 2.5 × 10^9^.	Barreto 2014 [128]: 24; 12 weeks; L. plantarum, milk (80 mL bottles × 1/day) 10^7^ CFU/g.
	Reininghaus 2020 [129]: MDD; 82; add-on; 4 weeks; B. bifidum W23, B. lactis W51, B. lactis W52, L. acidophilus W22, L. casei W56, L. paracasei W20, L. plantarum W62, L. salivarius W24 and Lactococcus lactis W19; 7.5 × 10^9^ CFU/day.	Bernini 2016 [130]: 51; 6 weeks; B. lactis HN019, milk (80 mL bottle × 1/day) 3.4 × 10^8^ CFU/mL.
	Majeed, 2018 [131]: MDD in IBS; 40; stand-alone; 90 days; Bacillus coagulans 2 × 10^9^	Cicero et al. 2020 [132] MetS elderly patients; 60 days; L. plantarum PBS067, L. acidophilus PBS066 and L. reuteri PBS072 (plus prebiotic).
	Bambling et al. 2017 [133]: TRD; 12; add-on; 8 weeks; L. acidophilus, B. bifidum, Streptoccocus thermophilus; 2 × 10^10^ CFU/day.	Behrouz et al. 2020 [134]: NAFLD; 71; add-on; 12 weeks; L. casei, L. rhamnosus, L. acidophilus, B. longum, and B. breve; 5 × 10^9^ CFU/day.
	Pinto-Sanchez et al. 2017 [135]: depression/anxiety in IBS; 44; 6 weeks; B. longum NCC 3001; 3 × 10^9^ CFU	Abhari et al. 2020 [136]: NAFLD; 53; 12 weeks; Bacillus coagulans (GBI-30) 10^9^ spore/day (plus inulin).
	Browne et al. 2021 [137]: depression/anxiety in pregnancy; stand-alone; 40; 8 weeks; B. bifidum W23, B. lactis W51, B. lactis W52, L. acidophilus W37, L. brevis W63, L. casei W56, L. salivarius W24, Lactococcus lactis W19 and Lactococcus lactis W58; 5 × 10^9^ CFU/day	Scorletti et al. 2020 [138]: NAFLD; 104; stand-alone; 10–14 months; fructo-oligosaccharides, 4 g twice per day, plus B. animalis subsp. lactis BB-12 10 × 10^9^ CFU/day (plus fructo-oligosaccharides).

Abbreviations: B.: Bifidobacterium; CFU: colony-forming unit; IBS: irritable bowel syndrome; L.: Lactobacillus; MDD: major depressive disorder; MetS: metabolic syndrome; NAFLD: non-alcoholic fatty liver disease; TRD: treatment-resistant depression.

**Table 4 pharmaceuticals-14-00384-t004:** Proposed key points of a randomized clinical trial protocol of probiotics efficacy in depressive patients with metabolic abnormalities.

Population	Depressive Disorders with Comorbid Obesity/MetS/NAFLD
Probiotics	Lactobacillus and Bifidobacterium strains mixture
Probiotic dose per day	min. 10^9^ CFU/day
Formulation	capsule
Intervention period	8 weeks
Primary outcome	depressive symptoms
Secondary outcomes	anthropometric parameters, MetS criteria, indicators of liver fibrosis, fecal microbiota composition and function analysis
Tertiary outcomes	cortisol, inflammation and oxidative stress parameters

## Data Availability

Data sharing not applicable.

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
