# Peer review of "Probiotics as a Treatment for “Metabolic Depression”? A Rationale for Future Studies"

_pharmaceuticals, 2021, doi:10.3390/ph14040384_

Round 1
Reviewer 1 Report
Having read the manuscript entitled „Probiotics as a treatment for “metabolic depression”? A rationale for future studies” I have to admit that the subject is interesting and important, whereas the manuscript is informative and well-written. Therefore I have only some minor suggestions before publication:
- A paragraph describing suggested mechanisms of action of probiotics in relation to "metabolic depression" should be added.
- The most important data from pre-clinical studies should be presented in a table.
Author Response
Dear Reviewer,
- A paragraph describing suggested mechanisms of action of probiotics in relation to "metabolic depression" should be added.
- We added the summarative information in lines 347-349.
- The most important data from pre-clinical studies should be presented in a table.
- We added a summarative table (Tabl. 1) in lines 151-153.
Thank you for your remarks,
Authors.
Reviewer 2 Report
This paper describes in-depth analysis of the latest literature on the treatment of metabolic diseases often coexisting with depression / anxiety. In addition, the Authors focused on a very frequently discussed problem in the medical world - the gut-brain axis. In recent years, there has been much interest in the role of microbiota changes (dysbiosis) in the development of chronic inflammation and civilization diseases. The review is divided into three sections: experimental studies, clinical studies and practical applications. There are many pertinent conclusions that indicate the limitations of the cited research e.g lines 314-316, lines 320-322. In the practical applications chapter, the Authors provide important tips for planning a clinical trial with probiotics. Overall this is a well-designed review and provides very interesting conclusions/suggestions.
With some minor editing to the manuscript this paper is deserving of publication.
There are many abbreviations in the paper. It is hard to read some parts of the manuscript and follow the author's thoughts. I recommend to delete some abbreviations e.g. LPS, DD, GDM. Especially since the authors use some abbreviations only once.
RCTs should be explained in line 155, where is is apeared for the first time.
Author Response
Dear Reviewer,
- There are many abbreviations in the paper. It is hard to read some parts of the manuscript and follow the author's thoughts. I recommend to delete some abbreviations e.g. LPS, DD, GDM. Especially since the authors use some abbreviations only once.
- We deleted some of the abbreviations: LPS, DD, TRD, GDM, BAT, WAT, CLGI.
- RCTs should be explained in line 155, where is is apeared for the first time.
- We expalined it.
Thank you for your remarks,
Authors.